# Molecular Mechanisms Underlying the Anti-Inflammatory Properties of (R)-(-)-Carvone: Potential Roles of JNK1, Nrf2 and NF-κB

**DOI:** 10.3390/pharmaceutics15010249

**Published:** 2023-01-11

**Authors:** Cátia Sousa, Bruno Miguel Neves, Alcino Jorge Leitão, Alexandrina Ferreira Mendes

**Affiliations:** 1Center for Neuroscience and Cell Biology, University of Coimbra, 3004-504 Coimbra, Portugal; 2Faculty of Pharmacy, University of Coimbra, 3000-548 Coimbra, Portugal; 3Centre for Innovative Biomedicine and Biotechnology, University of Coimbra, 3004-504 Coimbra, Portugal; 4Department of Medical Sciences and Institute of Biomedicine––iBiMED, University of Aveiro, 3810-193 Aveiro, Portugal

**Keywords:** (R)-(-)-carvone, inflammation, aging, NF-κB, Nrf2, MAPKs

## Abstract

To explore the molecular mechanisms underlying the anti-inflammatory activity of (R)-(-)-carvone, we evaluated its ability to inhibit the signaling pathways involving the mitogen-activated protein kinases (MAPKs) and the transcription factor, nuclear factor kappa-light-chain-enhancer of activated B cells (NF-κB). (R)-(-)-carvone significantly decreased c-Jun N-terminal kinase (JNK) 1phosphorylation, but not that of the other MAPKs, induced by bacterial lipopolysaccharides (LPS) in the RAW 264.7 macrophage cell line. Although (R)-(-)-carvone significantly inhibited resynthesis of the inhibitor of NF-κB (IκB)-α induced by LPS, it did not interfere with the canonical NF-κB activation pathway, suggesting that it may interfere with its transcriptional activity. (R)-(-)-carvone also showed a tendency to decrease the levels of acetylated NF-κB/p65 in the nucleus, without affecting the activity and protein levels of Sirtuin-1, the major NF-κB/p65 deacetylating enzyme. Interestingly, the nuclear protein levels of the transcription factor, nuclear factor (erythroid-derived 2)-like 2 (Nrf2) and the expression of its target,, heme oxygenase-1 (HO-1), an antioxidant enzyme, also showed a tendency to increase in the presence of (R)-(-)-carvone. Taken together, these results suggest that the ability of (R)-(-)-carvone to inhibit JNK1 and to activate Nrf2 can underlie its capacity to inhibit the transcriptional activity of NF-κB and the expression of its target genes. This study highlights the diversity of molecular mechanisms that can be involved in the anti-inflammatory activity of monoterpenes.

## 1. Introduction

Natural products are an enriched source of compounds with a variety of chemical structures and pharmacological activities. Anti-inflammatory activity is among the most studied and reported for natural products and their components [1]. Monoterpenes are a well-known class of natural compounds with anti-inflammatory activity [2,3] that show promise as potential therapies for numerous diseases, namely chronic age-related diseases, including neurodegenerative, metabolic, cardiovascular, renal, respiratory and musculoskeletal diseases that are characterized by a chronic, low-grade inflammatory state [4,5]. This anti-inflammatory activity results in most cases from inhibition of one or both of two critical signaling pathways that orchestrate the inflammatory response and include members of the mitogen-activated protein kinase (MAPK) family and the transcription factor, the nuclear factor kappa-light-chain-enhancer of activated B cells (NF-κB) [6,7,8]. Nonetheless, significant differences have been reported, with distinct compounds presenting different abilities to inhibit each one of those signaling pathways, even in the same cell type [7,8,9]. However, considering the large number of studies that report the anti-inflammatory activity of monoterpene compounds, studies that provide mechanistic insights regarding that activity are scarce. Such mechanistic elucidation is important in assessing therapeutic potential as different signaling pathways and intermediates can play distinct roles in diverse diseases, some being beneficial and others detrimental.

In our previous work, a standardized screening of the anti-inflammatory activity of selected monoterpene compounds having in common the *p*-menthane nucleus identified nine compounds with anti-inflammatory activity and allowed the recognition of structure–activity relationships. Among the active compounds, (S)-(+)-carvone and (R)-(-)-carvone were the most potent, inhibiting the expression of pro-inflammatory mediators [e.g., inducible nitric oxide synthase (NOS2) and interleukin-1β (IL-1β)] induced by lipopolysaccharides (LPS) in murine macrophages [10]. Subsequently, we started exploring the molecular mechanism underlying the anti-inflammatory effects of the two carvone enantiomers and found that they were dissimilar. Thus, we first focused on (S)-(+)-carvone and found that its anti-inflammatory activity is mediated, at least in part, by the activation of Sirtuin-1 (SIRT1) [11]. In the current study, we followed a similar approach to explore the molecular mechanisms involved in the anti-inflammatory activity of (R)-(-)-carvone (Figure 1). For that and considering the importance of the MAPKs and NF-κB in orchestrating the inflammatory response by inducing inflammatory gene expression and the role of SIRT1 in counteracting NF-κB transcriptional activity, we evaluated the ability of (R)-(-)-carvone to modulate these signaling pathways. As the results obtained suggest that (R)-(-)-carvone negatively modulates NF-κB transcriptional activity without interfering with its canonical activation pathway, but, unlike (S)-(+)-carvone, does not affect SIRT1 expression or activity, we explored other potential mechanisms.

## 2. Materials and Methods

### 2.1. Cell Culture and Treatments

The mouse macrophage cell line, RAW 264.7 (ATCC No. TIB-71), was cultured in DMEM supplemented with 10% non-heat-inactivated fetal bovine serum, 100 U/mL penicillin and 100 µg/mL streptomycin. RAW 264.7 cells were plated at a density of 3 × 10^5^ cells/mL and left to stabilize for up to 24 h. The cells were used between passages 25 and 40.

For cell treatments, (R)-(-)-carvone (#124931, purity 98%, Sigma-Aldrich Co., St Louis, MO, USA) was dissolved in dimethyl sulfoxide (DMSO; Sigma-Aldrich Co.). LPS from *Escherichia coli* 026:B6 (Sigma-Aldrich Co.) were dissolved in phosphate buffered saline (PBS). Concentrations of (R)-(-)-carvone were selected based on our previous work [10]. DMSO was used as a vehicle and added to the control and LPS-treated cell cultures so that in all conditions its final concentration did not exceed 0.1% (*v*/*v*). The chemicals used or the vehicle were added to macrophage cell cultures 1 h before the pro-inflammatory stimulus, 1 µg/mL LPS, and maintained for the rest of the experimental period. The concentrations of (R)-(-)-carvone and the experimental treatment periods are indicated in figures and/or figure legends.

### 2.2. Preparation of Cell Extracts

For total cell extracts preparation, macrophages monolayer culture was washed with ice-cold PBS, pH 7.4 and lysed with ice-cold RIPA buffer [150 mM sodium chloride, 50 mM Tris (pH 7.5), 5 mM ethylene glycol-bis(2-aminoethylether)-N,N,N0,N0-tetraacetic acid, 0.5% sodium deoxycholate, 0.1% sodium dodecyl sulfate, 1% Triton X-100, protease inhibitor cocktail (Complete,Mini, Roche Diagnostics, Mannheim, Germany)] and phosphatase inhibitor cocktail (PhosSTOP, Roche Diagnostics, Mannheim, Germany) and incubated on ice for 30 min. After this period, the lysates were centrifuged at 14,000 rpm for 10 min at 4 °C and the supernatants were stored at 20 °C until use. A Nuclear Extract Kit (Active Motif, La Hulpe, Belgium) was used for the preparation of cytoplasmic and nuclear extracts, following the manufacturer’s instructions. Protein concentration in the extracts was determined with the bicinchoninic acid kit (Sigma-Aldrich Co.).

### 2.3. Western Blot

Western blot was performed as described previously [12]. Briefly, total (25 µg), cytoplasmic (25 µg) or nuclear (30 µg) proteins were separated by SDS-PAGE under reducing conditions and electrotransferred onto PVDF membranes which were probed overnight at 4 °C or for 2 h at room temperature with the primary antibodies listed in Table 1 and then with anti-rabbit (dilution 1:20000; NIF1317, lot9465473, GE Healthcare, Chalfont St. Giles, UK) or anti-mouse (dilution 1:20000; NIF1316, lot6963606, GE Healthcare, Chalfont St. Giles, UK) alkaline phosphatase-conjugated secondary antibodies. Mouse monoclonal anti-β-tubulin I and rabbit polyclonal anti-lamin B1 were used as loading controls of total and cytoplasmic extracts and of nuclear extracts, respectively. Immune complexes were detected with Enhanced ChemiFluorescence reagent (GE Healthcare) in the imaging system Thyphoon^TM^ FLA 9000 (GE Healthcare). Image analysis was performed with TotalLab TL120 software (Nonlinear Dynamics Ltd., Newcastle upon Tyne, UK).

### 2.4. Immunocytochemistry

To evaluate the NF-κB/p65 activation, we assessed its nuclear translocation by immunocytochemistry, as described previously [11]. µ-Slide 8 Well chamber plates (ibiTreat, Ibidi, Martinsried, Germany) suitable for cell culture were used for macrophage culture. The cells were treated as indicated in figure legends. After washing with ice-cold PBS, pH 7.4, the cells were fixed in 4% paraformaldehyde at room temperature for 15 min. After three 5 min washes in 0.1 M glycine in PBS pH 7.4, the plates were blocked in 5% Goat Serum, 0.3% Triton in PBS, pH 7.4, for 1 h at room temperature, followed by overnight incubation, at 4 °C, with a rabbit monoclonal anti-NF-κB p65 (D14E12) XP^(R)^ antibody (dilution 1:400 in 1% Bovine Serum Albumin in PBS, pH 7.4; #8242S, Lot 4, Cell Signaling Technology, Inc.). Then, the plates were incubated for 1 h with the secondary antibody, anti-rabbit IgG (H + L) CF^TM^488A (dilution 1:400; SAB4600165, Lot 10C0615, Biothium, Inc., Fremont, CA, USA). DAPI (0.2 ng/mL; Molecular Probes, Invitrogen, Eugene, OR, USA) was used to counterstain the nuclei. Specificity was confirmed in negative controls set up by omitting the primary antibody. Images were acquired in an Axio Observer ZI fluorescence microscope (Carl Zeiss, Germany).

### 2.5. SIRT1 Activity Assay

The ability of (R)-(-)-carvone to activate SIRT1 was evaluated using the SIRT1 Direct Florescent Screening Assay Kit (Cayman Chemical Company, Ann Arbor, MI, USA) following the manufacturer’s instructions. Briefly, as described in our previous study [11], the kit is based on the deacetylation of a p53-derived peptide conjugated with a fluorophore, by recombinant human SIRT1 in the presence of its co-factor, NAD^+^. The deacetylation reaction yields a fluorescent product whose fluorescence intensity is directly proportional to the enzyme activity, which can be increased or decreased in the presence of activating or inhibiting compounds, respectively. Results are presented as mean fluorescence intensity (arbitrary units) ± SEM. 

### 2.6. Statistical Analysis

Results are presented as means ± SEM. Statistical analysis was performed with GraphPad Prism version 6.0 (GraphPad Software, San Diego, CA, USA), using one-way ANOVA with the Dunnett post-test to compare multiple conditions to a control group. In Figure 6B, the Mann–Whitney test was used to assess the statistical significance of the differences between each condition and the basal SIRT1 activity, as those results did not follow a normal distribution. Results were considered statistically significant at *p* < 0.05.

## 3. Results

### 3.1.(R)-(-)-Carvone Inhibits LPS-Induced Phosphorylation of JNK1, but Not That of Other JNK Isoforms, p38 and ERK1/2

Upon binding to its receptor, the toll-like receptor 4 (TLR4), LPS triggers the activation of all MAPK family members, namely extracellular signal-regulated kinase (ERK)1/2, p38 and c-Jun N terminal kinase (JNK) by inducing their phosphorylation on specific serine/threonine or tyrosine residues [13]. 

We observed previously that treatment of the RAW 264.7 cell line with LPS for 5 min is sufficient to significantly increase p38 and JNK phosphorylation in comparison to control cells, while ERK1/2 phosphorylation is maximal after 1 h [11]. Using these time points, pre-treatment with (R)-(-)-carvone did not significantly affect LPS-induced p38 (Figure 2A) and ERK1/2 (Figure 2B) phosphorylation. On the opposite, (R)-(-)-carvone significantly decreased LPS-induced phosphorylation of JNK1 (Figure 2C), while the other two isoforms, JNK2 and 3, were not significantly affected, although a tendency to reduced levels of the activated forms can be observed (Figure 2C). Taken together, these results show that (R)-(-)-carvone efficiently inhibits JNK1 activation, but does not affect LPS-induced activation of the other MAPKs, including its isoforms, JNK2 and 3.

### 3.2.(R)-(-)-Carvone Does Not Interfere with the Canonical Activation Pathway and Nuclear Translocation of NF-κB

In the canonical NF-κB activation pathway, recognition of a suitable agonist (e.g., LPS) by its receptor (e.g., TLR4) triggers a signaling cascade that leads to the phosphorylation of the IκB kinase complex (IKK), in particular of its catalytic subunit, IKKβ [14]. Then, IKKβ phosphorylates IκBα, the natural inhibitor of NF-κB that retains its dimers in the cytoplasm. Phosphorylation of IκBα at serine (Ser) 32 and 36 triggers its ubiquitination and subsequent proteasomal degradation [14]. Thus, NF-κB dimers, typically composed of p65 or RelA and p50 proteins, are freed and translocate to the nucleus [14]. Since IκBα phosphorylation and degradation are key steps in the canonical NF-κB activation pathway, the effect of (R)-(-)-carvone on these processes was evaluated. As expected, LPS induced IκBα phosphorylation (Figure 3A) and its subsequent degradation (Figure 3B), but (R)-(-)-carvone did not interfere with any of these steps (Figure 3A,B). 

Then, to determine whether (R)-(-)-carvone can interfere with the nuclear translocation of the NF-κB dimers, an immunocytochemistry assay for NF-κB/p65 was performed. Figure 4A shows that NF-κB/p65 is located in the cytoplasm in vehicle-treated control cells, while after treatment with LPS for 20 min, both in the presence and absence of (R)-(-)-carvone, immunoreactivity is mostly located in the nucleus.

Further confirming these results, Western blot analyses show that LPS decreased the cytoplasmic (Figure 4B, left side) and increased the nuclear (Figure 4B, right side) levels of NF-κB/p65, which were not modified by pre-treatment with (R)-(-)-carvone. 

Besides release from complexes with IκB-α and subsequent nuclear translocation, full NF-κB transcriptional activity requires the NF-κB/p65 subunit to undergo various post-translation modifications (PTMs) [15]. Phosphorylation of NF-κB/p65 on Ser536 by IKKβ is one of the PTMs essential for a full NF-κB transcriptional activity [16,17,18]. Figure 4C shows that pre-treatment with (R)-(-)-carvone had no effect on the phosphorylation levels of NF-κB/p65 at Ser536 induced by LPS (Figure 4C). 

Taken together, these results show that (R)-(-)-carvone does not interfere with the canonical NF-κB activation pathway, including its nuclear translocation and phosphorylation on Ser536.

### 3.3. (R)-(-)-Carvone Inhibits IκB-α Resynthesis

Our previous results demonstrated that (R)-(-)-carvone decreased the expression of two NF-κB target genes, NOS2 and IL-1β [10]. However, the results above showed that this compound did not interfere with the canonical NF-κB activation pathway. Thus, to further confirm those previous results, the expression of another NF-κB-dependent gene, IκB-α [19], was evaluated. For this, we first performed time course experiments to determine the time point after the addition of the inflammatory stimulus, at which the cytoplasmic protein levels of IκB-α started to increase after having been degraded. We observed previously that newly synthesized IκBα was detectable 20 min after the addition of LPS to macrophage cultures, reaching the maximum at 60 min [11]. Thus, the effect of (R)-(-)-carvone on IκB-α resynthesis was assessed 60 min after the addition of LPS. Figure 5 shows that pre-treatment of macrophage cultures with (R)-(-)-carvone significantly reduced the increase in IκBα protein levels induced by LPS. This result further suggests that (R)-(-)-carvone interferes with NF-κB transcriptional activity, likely by an indirect mechanism that does not involve its canonical activation pathway.

### 3.4. (R)-(-)-Carvone Tends to Decrease LPS-Induced Acetylation of NF-κB/p65 at Lys310 Independently of SIRT1 Activity and Expression

The results presented above confirm that, despite not interfering with the canonical NF-κB activation pathway, (R)-(-)-carvone is likely to disturb its transcriptional activity. Thus, we hypothesized that the underlying mechanism may involve inhibition of activating the PTMs of the NF-κB/p65 subunit, other than Ser536 phosphorylation already shown not to be affected by (R)-(-)-carvone (Figure 4C). Among these PTMs, acetylation of lysine (Lys) 310 on NF-κB/p65 is especially relevant, being required for its full transcriptional activity [20]. Figure 6A shows that treatment with LPS slightly increased the levels of NF-κB/p65 acetylated at Lys310 (Ac-NF-κB/p65), although the difference is not statistically significant. (R)-(-)-carvone decreased Ac-NF-κB/p65 levels to those obtained in the control or even less, in all experiments, suggesting that the effect induced by LPS, although small, is real. The differences, in either case, did not reach statistical significance, likely due to the small magnitude of the effect induced by LPS. 

Changes in the levels of acetylated NF-κB/p65 can be caused either by activation of acetylating enzymes, namely CBP/p300, or by activation of deacetylating enzymes. Among these, SIRT1, a NAD^+^-dependent class III histone/protein deacetylase (HDAC), is especially important as it physically interacts with NF-κB/p65, specifically deacetylating its Lys310 residue, and, consequently, inhibiting its transcriptional activity [21]. Thus, we next evaluated the ability of (R)-(-)-carvone to modulate SIRT1 activity and/or abundance. Figure 6B shows that (R)-(-)-carvone was unable to modify the basal activity of SIRT1 towards a specific acetylated substrate. Furthermore, (R)-(-)-carvone also did not affect SIRT1 protein levels when compared to cells treated with LPS alone (Figure 6C). Altogether, these results point out that induction of SIRT1 protein levels and/or activity is unlike to be the mechanism responsible for the inhibitory effect of (R)-(-)-carvone on LPS-induced NF-κB-dependent gene transcription.

### 3.5. (R)-(-)-Carvone Promotes Nrf2 Nuclear Translocation and the Expression of its Target Gene, Heme Oxygenase-1

As mentioned above, CBP/p300 is a co-activator and histone/protein acetyltransferase essential for full NF-κB transcriptional activity by acetylating NF-κB/p65 on Lys310, without interfering with its DNA binding [18]. Since (R)-(-)-carvone showed a tendency to decrease Lys310 acetylation without affecting SIRT1 protein levels or activity, we hypothesized that it may instead interfere with CBP/p300. Indeed, the availability of CBP/p300 for interaction with NF-κB is limited by competition with the transcription factor, nuclear factor (erythroid-derived 2)-like 2 (Nrf2) [22]. As this transcription factor is activated by electrophilic compounds, which is the case of (R)-(-)-carvone due to the Michael center resulting from the conjugation of the carbonyl group and the α,β double bond [23,24], we hypothesized that this compound may activate Nrf2 and consequently decrease the availability of CBP/p300 for interaction with NF-κB [22]. Results in Figure 7A show that treatment with (R)-(-)-carvone alone for 30 and 60 min is sufficient to induce a mean increase of the nuclear levels of Nrf2 of approximately 1.8- and 3.6-fold relative to control cells. Accordingly, the protein levels of heme oxygenase-1 (HO-1), one of the most relevant Nrf2 target genes, were also substantially increased by more than 35-fold relative to control cells (Figure 7B). Moreover, LPS elicited small increases both in nuclear Nrf2 levels and HO-1 expression, which were much higher in cells pre-treated with (R)-(-)-carvone (Figure 7C,D). Although none of these results reached statistical significance due to inter-assay variability, the magnitude of the effects observed is impressive and the relative differences among the experimental groups were consistently maintained in all assays performed. Thus, such effects are unlikely due to random events, but rather to real effects of (R)-(-)-carvone.

## 4. Discussion

The results presented show that (R)-(-)-carvone inhibits JNK1 (Figure 2C) which contributes to the inflammatory response by several mechanisms. On one hand, JNKs phosphorylate c-Jun, which prevents its ubiquitination and subsequent degradation, increasing its transcriptional activity, particularly as a component of the transcription factor, activating Protein-1 (AP-1) [25,26]. This transcription factor, in turn, is associated with the induction of several inflammation-related genes [27,28]. Moreover, c-Jun phosphorylation induced by IL-1 was shown to be required for NF-κB/p65 recruitment to the *ccl2* gene which codes for C-C Motif Chemokine Ligand 2 (CCL2) or Monocyte Chemoattractant Protein-1 (MCP-1), a potent chemokine and inflammatory mediator [29]. Interestingly, this study also showed that JNK1/2-induced phosphorylation of c-Jun facilitates Histone Deacetylase 3 (HDAC3) dissociation from that gene while enhancing the recruitment of CBP/p300 and NF-κB to promote *ccl2* transcription [29]. Even though cell, stimulus and gene specificities can be involved, the ability of (R)-(-)-carvone to inhibit LPS-induced JNK1 activation is in line with that mechanism, whereby inhibition of c-Jun phosphorylation may impair NF-κB/p65 acetylation by CBP/p300 and its recruitment to target genes, thus inhibiting their expression. Further studies are required to confirm these mechanisms and that phosphorylated c-Jun plays the same role in modulating the expression of other NF-κB-dependent genes, such as IκB-α (Figure 5), NOS2 and IL-1β [10], in response to a different yet also pro-inflammatory and NF-κB-inducing stimulus. Nonetheless, considering that (R)-(-)-carvone had no effect on the phosphorylation of the JNK2 and 3 isoforms, the contribution of JNK1 inhibition to the anti-inflammatory effect of this compound may be limited. 

On the other hand, while acetylation of p65 on Lys310 by CBP/p300 can be promoted by JNKs [29], it can be inhibited by competition for this acetylase between NF-κB and Nrf2 [22] and can be reversed through deacetylation by SIRT1 [21]. While we observed no effects of (R)-(-)-carvone on SIRT1 protein levels and activity (Figure 6B,C), we found a tendency to decreased levels of p65 acetylated on Lys 310 in cells treated with LPS in the presence of (R)-(-)-carvone, relative to cells treated with LPS alone (Figure 6A). This suggests that inhibition of p65 acetylation, rather than induction of its deacetylation, is a mechanism relevant for the inhibitory effect of (R)-(-)-carvone on the expression of NF-κB target genes. Besides inhibition of JNK1 activation (Figure 2C), we found that (R)-(-)-carvone is sufficient to induce the nuclear translocation and accumulation of Nrf2 and the expression of its target gene, HO-1 (Figure 7). Thus, inhibition of JNK1 (Figure 2C) and activation of Nrf2 (Figure 7) may both contribute to the observed decreased expression of NF-κB target genes (Figure 5 and [10]) by interfering with p65 acetylation by CBP/p300, without affecting its canonical activation pathway (Figs. 3 and 4). This agrees with a recent study that reported the ability of (R)-(-)-carvone to inhibit NF-κB transcriptional activity induced by TNF-α in a human embryonic kidney cell line [30]. However, this study did not elucidate whether the inhibitory effect of (R)-(-)-carvone resulted from inhibition of the canonical activation pathway or occurred at the level of transcriptional activity.

Moreover, Nrf2 is also essential to activate antioxidant mechanisms, while increased production of reactive oxygen species is a characteristic feature of the inflammatory response, being required among other mechanisms for NF-κB [31] and AP-1 activation [32]. Thus, the ability of (R)-(-)-carvone to stabilize the Nrf2 protein and increase its nuclear levels and transcriptional activity (Figure 7) is another relevant mechanism by which it may decrease NF-κB activation and counteract the inflammatory response. As essential components of the inflammatory response and oxidative stress, MAPKs and NF-κB are critical molecular determinants of many diseases, particularly of those associated with aging [5,33]. Therefore, they are considered promising targets for the development of therapeutic strategies to halt or, at least, slow down the progression of those diseases. In this regard, many natural compounds have been found to have positive effects in *in vitro* and *in vivo* models of many diseases by inhibiting those processes [8,9,34] and/or activating anti-inflammatory pathways such as Nrf2 and SIRT1 [35,36,37]. The results obtained in this study suggest that (R)-(-)-carvone may be useful by targeting the interaction between inflammatory and anti-inflammatory/antioxidant signaling pathways and intermediates. Interestingly, some recent *in vivo* studies reported interesting activities of (R)-(-)-carvone, namely hypolipidaemic and insulin secretagogue activities [38], protection from doxorubicin-induced cardiotoxicity [39] and anti-allergic airway inflammation [40]. Although the molecular mechanisms underlying these activities were not clearly elucidated, the ability of (R)-(-)-carvone to activate Nrf2 and inhibit NF-κB may be especially relevant since all these conditions are associated with oxidative stress and inflammation. In one of these studies, for instance, (R)-(-)-carvone decreased catalase activity in the liver of hyperlipidaemic mice [38]. Since catalase is an antioxidant enzyme whose expression is induced by Nrf2 [41], it can be speculated that activation of this transcription factor by (R)-(-)-carvone is a relevant mechanism underlying its beneficial effects in hyperlipidemic mice. Osteoarthritis is another disease in which oxidative stress and inflammation play significant roles and for which no effective treatments exist. Reduction of those processes by natural and synthetic compounds that inhibit NF-κB and/or activate Nrf2 has been shown to decrease catabolic responses *in vitro* [8,42,43] and to reduce joint damage *in vivo* [44,45,46], suggesting that due to its effect on those transcription factors, (R)-(-)-carvone may also be effective for osteoarthritis treatment. This study has some limitations, namely the lack of statistical significance of some of the results obtained. As shown in the raw data presented in Appendix A, the results are consistent across replicates, although inter-assay variability is high and likely prevented statistical significance being achieved.

In summary, the results presented show that (R) -(-)-carvone inhibits JNK1 activation and activates Nrf2, which are relevant as anti-inflammatory mechanisms, by interfering with various signaling pathways, some of which converge on NF-κB activation. Among those pathways, inhibition of the interaction between NF-κB and CBP/p300 may be especially relevant for the anti-inflammatory activity of (R)-(-)-carvone, since that interaction is regulated in opposite ways by JNK and Nrf2 (Figure 8). Further studies, namely immunoprecipitation assays, are required to confirm that JNK1 inhibition and Nrf2 activation converge to reduce CBP/p300 binding to NF-κB/p65, thus underlying the anti-inflammatory effects of (R)-(-)-carvone. 

On the other hand, these findings are in contrast with the mechanism through which most monoterpene compounds have been reported to inhibit NF-κB activity, that is, inhibition of its canonical activation pathway, namely IκBα phosphorylation and degradation [7,47,48]. The ability of (R)-(-)-carvone to inhibit inflammation by inhibiting JNK1 and activating Nrf2, likely interfering in their crosstalk with NF-κB, highlights the diversity of molecular mechanisms that can underlie the anti-inflammatory properties of distinct monoterpenes. In this regard, (R)-(-)-carvone and its enantiomer, (S)-(+)-carvone [11] are especially interesting as both inhibit NF-κB-dependent inflammatory gene expression, yet through different mechanisms, none of which involve inhibition of IκBα phosphorylation and degradation. Considering that distinct pathways may contribute differentially to inflammation in diverse diseases, in-depth elucidation of the molecular mechanisms underlying the anti-inflammatory properties of different monoterpenes may lead to the development of targeted disease-specific therapies.

## Figures and Tables

**Figure 1 pharmaceutics-15-00249-f001:**
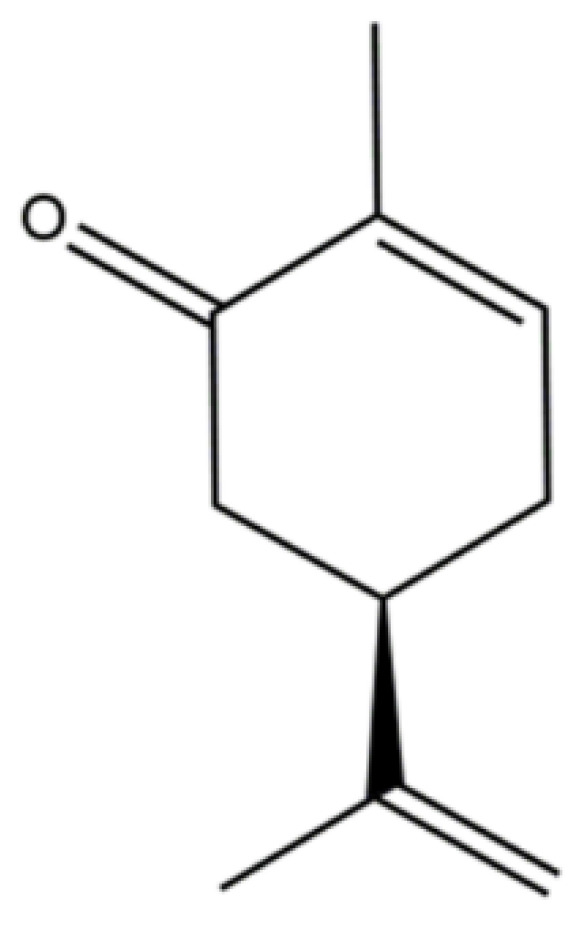
Structural formula of (R)-(-)-carvone.

**Figure 2 pharmaceutics-15-00249-f002:**
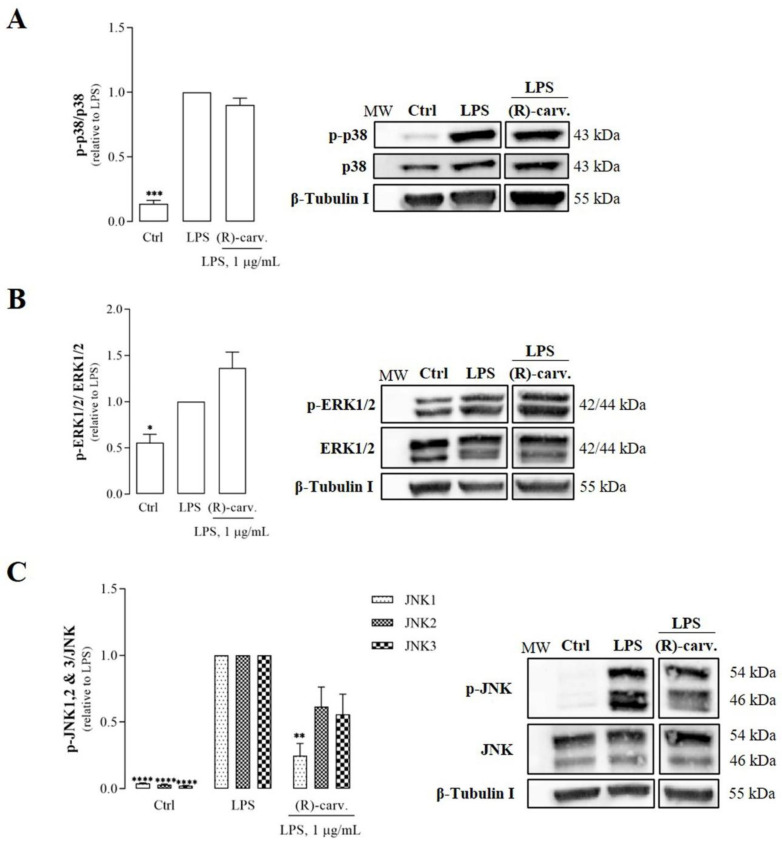
Effect of (R)-(-)-carvone on p38 (**A**), ERK1/2 (**B**) and JNK (**C**) phosphorylation induced by LPS in RAW 264.7 cells. Macrophage cultures were pre-treated with 665 µM of (R)-(-)-carvone or with the vehicle (0.1% DMSO) for 1 h and then with 1 µg/mL LPS for 5 min (**A**,**C**) or 1 h (**B**). Control cells (Ctrl) were treated with the vehicle (0.1% DMSO) in the absence of LPS. Each column represents the mean ± SEM of four (**A**,**C**) or three (**B**) independent experiments. Appendix A show the raw data of the results presented in panel C. Representative images are shown. The corresponding images of the full-length blots are shown in Appendix A. * *p* < 0.05,** *p* < 0.01, *** *p* < 0.001 and **** *p* < 0.0001 relative to LPS-treated cells. MW: molecular weight marker.

**Figure 3 pharmaceutics-15-00249-f003:**
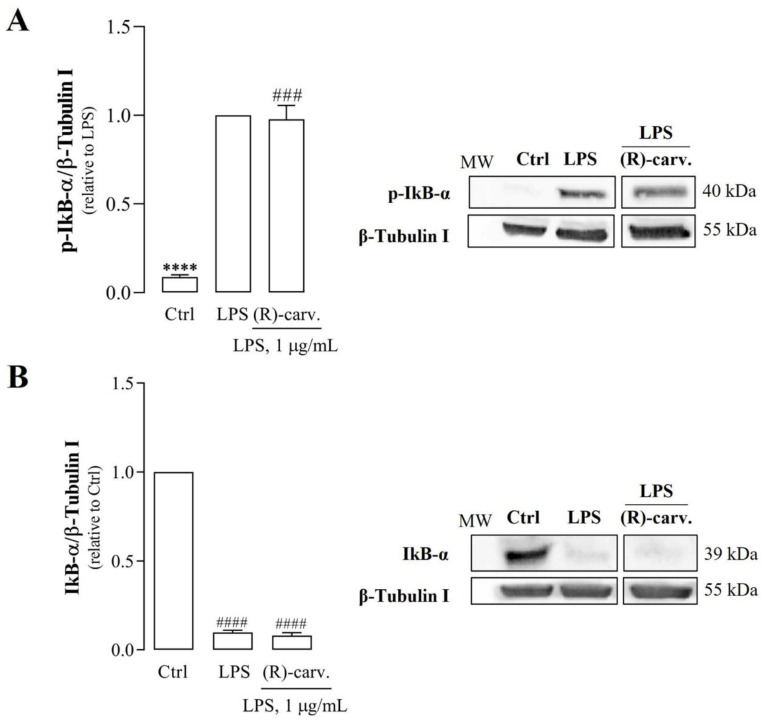
(R)-(-)-carvone does not affect the NF-κB activation pathway, namely, phosphorylation (**A**) and degradation (**B**) of IκB-α. Macrophage cultures were pre-treated with the vehicle (0.1% DMSO, Ctrl), or 665 µM (R)-(-)-carvone for 1 h, followed by treatment with 1 µg/mL LPS for 5 min (**A**) or 15 min (**B**). Each column represents the mean ± SEM of five (**A**) or four (**B**) independent experiments. Representative images are shown. The corresponding images of the full-length blots are shown in Appendix A. **** *p* < 0.0001 relative to LPS-treated cells. ^###^ *p* < 0.001 and ^####^ *p* < 0.0001 relative to the Ctrl. MW: molecular weight marker.

**Figure 4 pharmaceutics-15-00249-f004:**
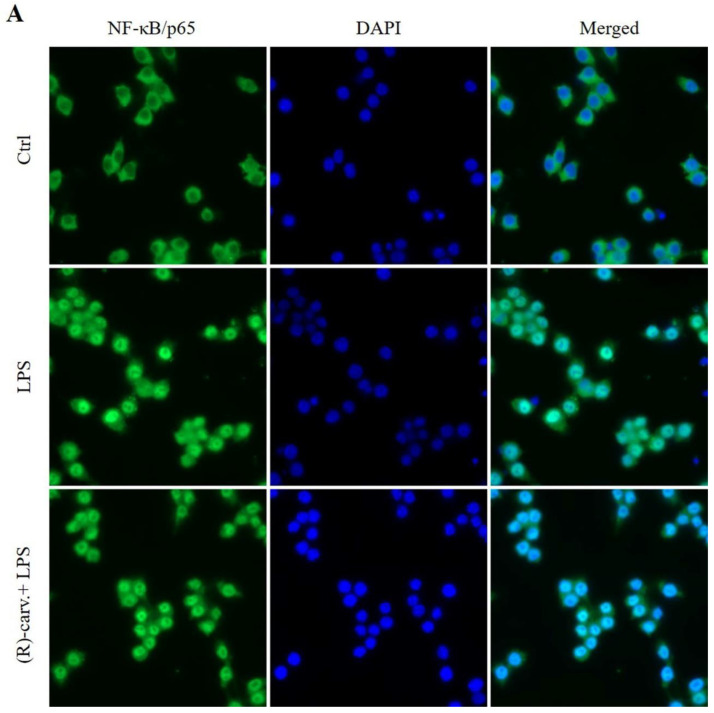
NF-κB/p65 nuclear translocation (**A**,**B**) and phosphorylation at Ser536 (**C**) are not inhibited by (R)-(-)-carvone. (**A**) Macrophages were pre-treated with the vehicle (0.1% DMSO) or 665 µM (R)-(-)-carvone for 1 h, and then with 1 µg/mL LPS for 20 min. Control cells (Ctrl) were treated with the vehicle (0.1% DMSO) alone. Immunofluorescence staining of NF-κB/p65 (green) and the nuclei (blue) were performed as described in Materials and Methods**.** Representative images of each condition are shown. Scale bar: 20 µm. (**B**,**C**) Macrophages were treated with 1 µg/mL LPS for 1 h (**B**) or 15 min (**C**), following pre-treatment with the vehicle (0.1% DMSO) or 665 µM (R)-(-)-carvone for 1 h. Total (**B**) and Ser536-phosphorylated (**C**) RelA/p65 levels were evaluated by Western blot in cytoplasmic and nuclear extracts (**B**) and in total cell extracts (**C**). Control cells (Ctrl) were treated with the vehicle alone (0.1% DMSO). Each column represents the mean ± SEM of four (cytoplasmic levels in panel (**B**) and panel (**C**) or six (nuclear levels in panel **B**) independent experiments. Representative images are shown. The corresponding images of the full-length blots are shown in Appendix A. *** *p* < 0.001 and **** *p* < 0.0001 relative to LPS-treated cells. ^##^ *p* < 0.01 relative to the Ctrl. MW: molecular weight marker.

**Figure 5 pharmaceutics-15-00249-f005:**
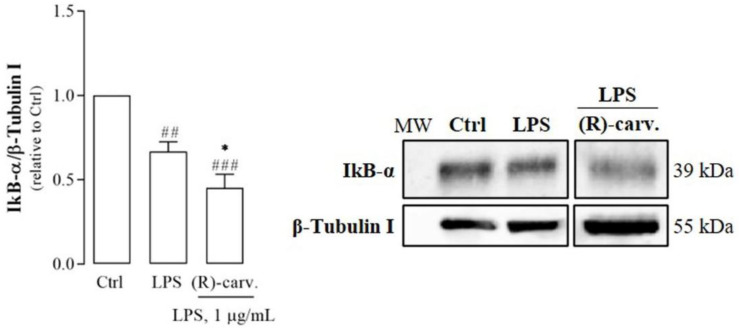
LPS-induced IκB-α resynthesis is inhibited by (R)-(-)-carvone. Cell cultures were pre-treated with the vehicle (0.1% DMSO) or 665 µM (R)-(-)-carvone for 1 h, followed by stimulation with 1 µg/mL LPS for 1 h. Control cells (Ctrl) were treated with the vehicle (0.1% DMSO) alone. Each column represents the mean ± SEM of seven independent experiments. Representative images are shown. The corresponding images of the full-length blots are shown in Appendix A. * *p* < 0.05 relative to LPS-treated cells. ^##^ *p* < 0.01 and ^###^ *p* < 0.001 relative to the Ctrl. MW: molecular weight marker.

**Figure 6 pharmaceutics-15-00249-f006:**
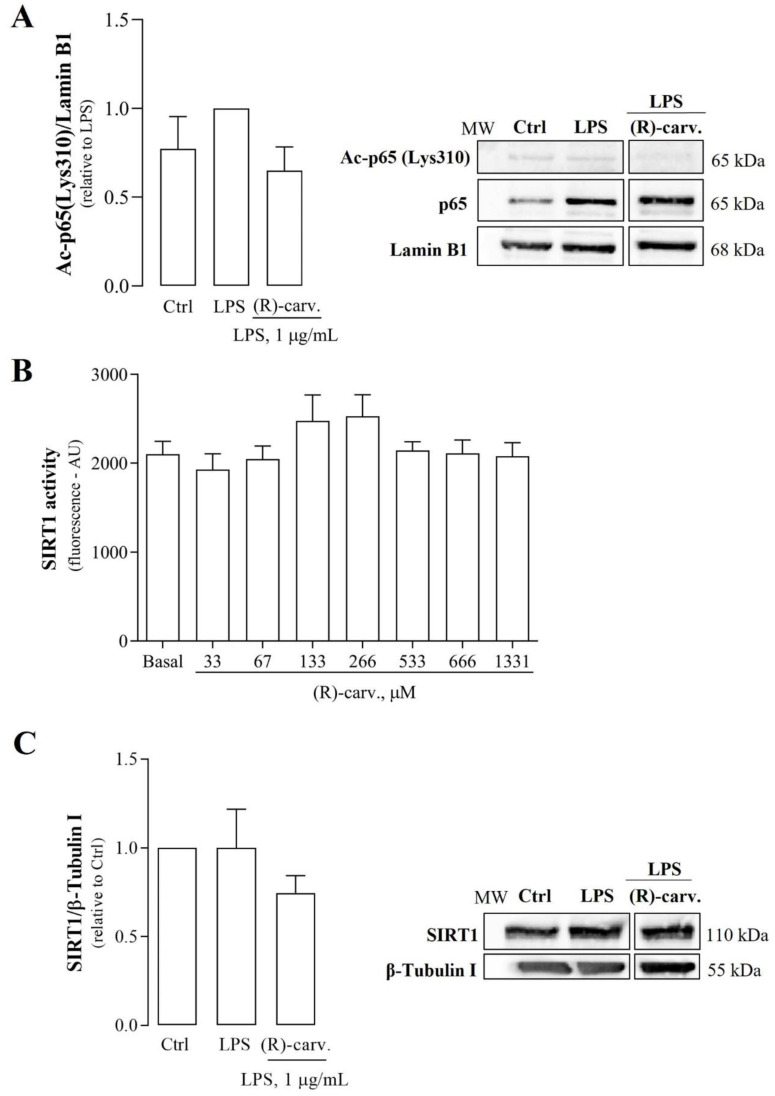
Effect of (R)-(-)-carvone on acetylated Lys310 NF-κB/p65 (Ac-p65) levels (**A**), SIRT1 activity (**B**) and protein levels (**C**). In (**A**,**C**), macrophages were pre-treated with the vehicle (0.1% DMSO) or 665 µM (R)-(-)-carvone for 1 h, followed by treatment with 1µg/mL LPS for 1 h. Control cells (Ctrl) were treated with the vehicle (0.1% DMSO) alone. (**B**) Different concentrations of (R)-(-)-carvone were directly incubated with human recombinant SIRT1 and a specific substrate, as described in Materials and Methods. Each column represents the mean ± SEM of three(**A**), at least six, (**B**) and three (**C**) independent experiments. Appendix A shows the raw data of the results presented in panel A. Representative images are shown. The corresponding images of the full-length blots are shown in Appendix A. MW: molecular weight marker.

**Figure 7 pharmaceutics-15-00249-f007:**
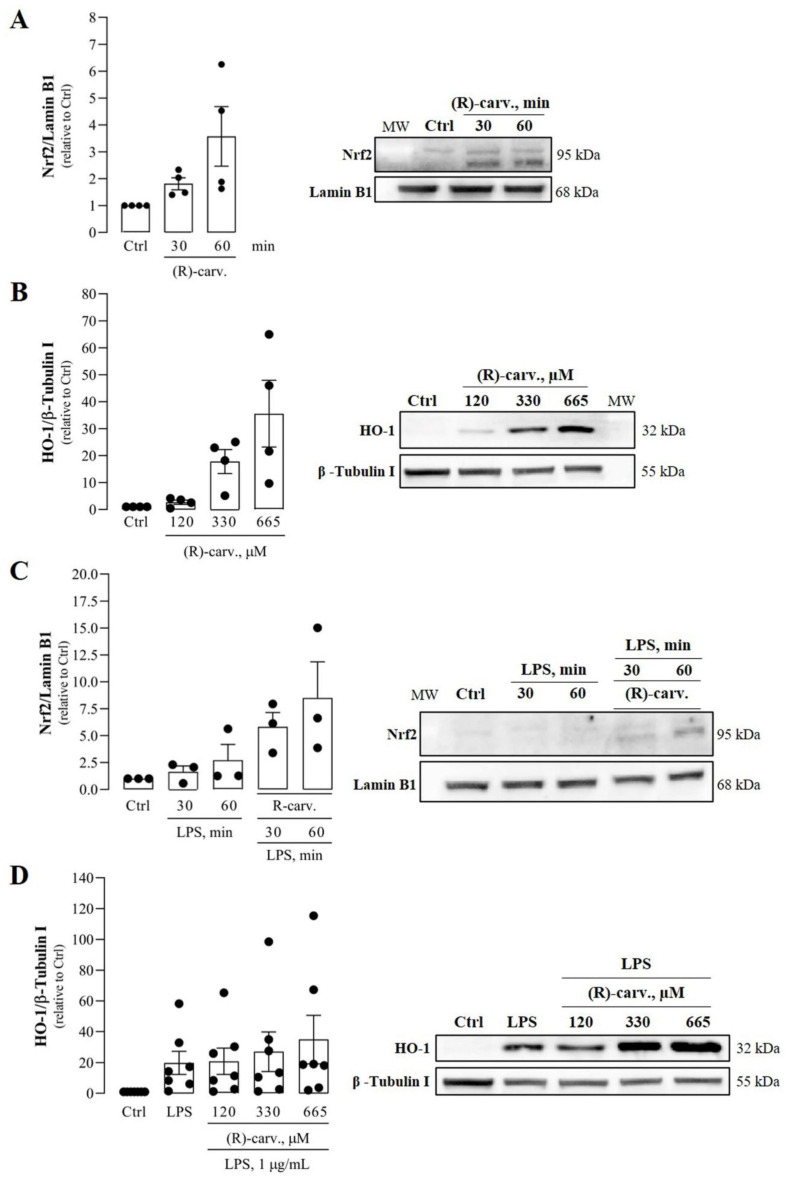
Effect of (R)-(-)-carvone on nuclear Nrf2 and Heme-oxygenase 1 (HO-1) levels in the absence (**A**,**B**) and presence (**C**,**D**) of LPS. In (**A**,**B**)**,** macrophages were treated with the vehicle (0.1% DMSO), 665 µM or the indicated concentrations of (R)-(-)-carvone for the time periods indicated (**A**) or for 18 h (**C**). In (**C**,**D**), macrophages were pre-treated with the vehicle (0.1% DMSO), 665 µM or the indicated concentrations of (R)-(-)-carvone for 1 h, followed by the addition of 1 μg/mL LPS for the time periods indicated (**B**) or for18 h (**D**). In all panels, control cells (Ctrl) were treated with the vehicle (0.1% DMSO) alone. Each column represents the mean ± SEM of four (**A**,**B**), three (**C**) or seven (**D**) independent experiments. Appendix A show the raw data of the results presented in each panel. Representative images are shown. The corresponding images of the full-length blots are shown in Appendix A. MW: molecular weight marker.

**Figure 8 pharmaceutics-15-00249-f008:**
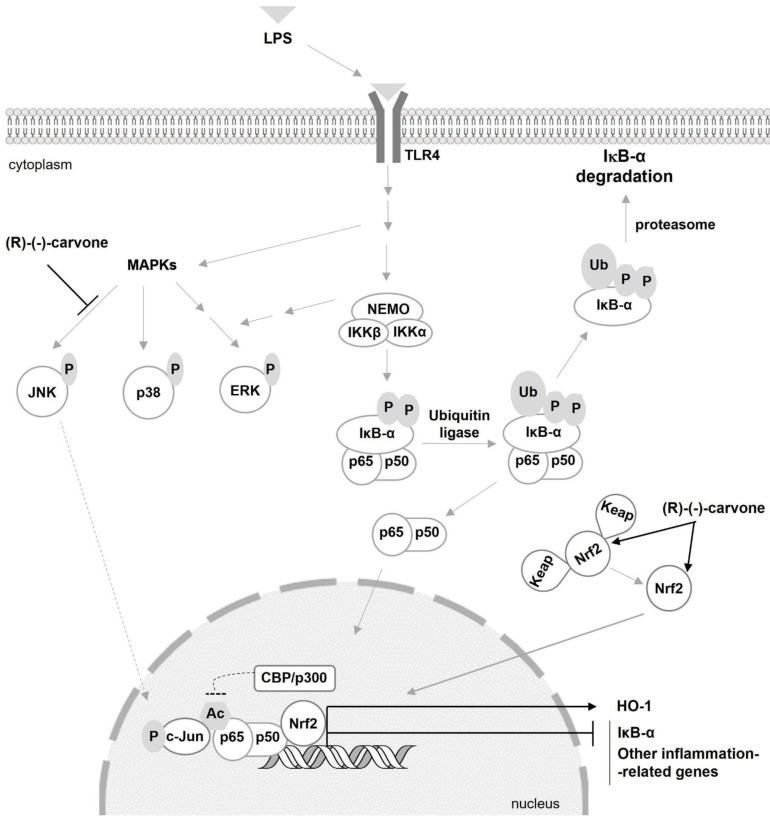
The proposed mechanism of action of (R)-(-)-carvone. (R)-(-)-carvone inhibits LPS-induced JNK phosphorylation, in particularly isoform 1, and activates Nrf2, both of which interfere with the acetylation of NF-κB subunit, p65, at Lys310, consequently decreasing the expression of IκB-α and other inflammation-related genes. The activation of Nrf2 by (R)-(-)-carvone increases Hemoxygenase-1 (HO-1) expression. Ac: acetyl; P––phosphate; Ub: ubiquitin.

**Table 1 pharmaceutics-15-00249-t001:** List of primary antibodies used in Western blot.

Protein	Source	Clonality	Dilution	Supplier	Catalogue/Lot Number
phospho-p44/42 MAPK (ERK1/2) (Thr202/Tyr204)	rabbit	polyclonal	1:1000	Cell Signaling Technology, Inc., Danvers, MA, USA	9101/27
p44/42 MAPK (ERK1/2)	rabbit	polyclonal	1:1000	Cell Signaling Technology, Inc.	9102/26
phospho-p38 MAPK (Thr180/Tyr182)	rabbit	polyclonal	1:1000	Cell Signaling Technology, Inc.	9211/ 21
p38 MAPK	rabbit	polyclonal	1:1000	Cell Signaling Technology, Inc.	9212/17
phospho-SAPK/JNK (Thr183/Tyr185)	rabbit	monoclonal	1:1000	Cell Signaling Technology, Inc.	4668/11
SAPK/JNK	rabbit	polyclonal	1:1000	Cell Signaling Technology, Inc.	9252/17
phospho-IκB-α (Ser32/36)	mouse	monoclonal	1:1000	Cell Signaling Technology, Inc.	9246/14
IκB-α	rabbit	polyclonal	1:1000	Cell Signaling Technology, Inc., Danvers, MA, USA	9242/9
NF-κB p65 (D14E12) XP^(R)^	rabbit	monoclonal	1:1000	Cell Signaling Technology, Inc.	8242/4
phospho- NF-κB p65 (Ser536)	rabbit	monoclonal	1:1000	Cell Signaling Technology, Inc.	3033/14
acetyl-NF-κB p65 (Lys310)	rabbit	polyclonal	1:750	Cell Signaling Technology, Inc.	3045/2
Sirtuin-1	rabbit	polyclonal	1:1000	Sigma-Aldrich Co.	07-131/2736563
Nrf2 (C-20)	rabbit	polyclonal	1:500	Santa Cruz Biotechnology, Inc., Dallas, TX, USA	sc-722/A1612
Heme oxygenase-1	mouse	monoclonal	1:1000	Invitrogen, Thermo Fisher Scientific, Waltham, MA, USA	MA 1-112/TK2665301
β-tubulin I	mouse	monoclonal	1:20,000	Sigma-Aldrich Co.	T7816/052M4835
lamin B1	rabbit	polyclonal	1:1000	Abcam, Cambridge, UK	ab16048/ GR48958-1

## Data Availability

Data is contained within the article or Appendix A.

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
