# Peer review of "Molecular Mechanisms Underlying the Anti-Inflammatory Properties of (R)-(-)-Carvone: Potential Roles of JNK1, Nrf2 and NF-κB"

_pharmaceutics, 2023, doi:10.3390/pharmaceutics15010249_

Round 1
Reviewer 1 Report
Authors showed that the ability of (R)-(-)-carvone to inhibit the phosphorylation of JNK1 and to activate Nrf2 transcriptional activity can underlie its capacity to inhibit the transcriptional activity of NF-kB. Although I think that it is a well-developed and interesting study, it is necessary to consider the following contents.
(1) In Figs. 6 and 7, authors speculated that (R)-(-)-carvone may activate Nrf2 and consequently decrease the availability of CBP/p300 and NF-kB interaction and NF-kB acetylation by CBP/p300. However, whether (R)-(-)-carvone inhibits the interaction between NF-kB and CBP/p300 has only been indirectly investigated. Direct proof should be provided by immunoprecipitation as to whether (R)-(-)-carvone inhibits the interaction.
(2) A model diagram of the molecular mechanism of (R)-(-)-carvone obtained in this paper would help the reader to better understand it.
Author Response
Coimbra, December 30th, 2022
Dear Editor and Reviewers:
On behalf of all authors, I thank you for the constructive comments and suggestions and for the opportunity to revise our manuscript. We think that the changes introduced greatly improved the manuscript and thank you again for your contribution.
As detailed below, we made significant changes to the text and some figures and also included additional data as supplementary tables to make clear how we reached the conclusions presented.
We do hope that you find the new version suitable for publication in this special issue of Pharmaceutics and remain at your disposal for any further corrections.
Kind regards and our best wishes for the new year,
Alexandrina Ferreira Mendes
Response to Reviewer 1
We thank the reviewer for positive comments and constructive suggestions. Please, find our responses beneath each point.
Authors showed that the ability of (R)-(-)-carvone to inhibit the phosphorylation of JNK1 and to activate Nrf2 transcriptional activity can underlie its capacity to inhibit the transcriptional activity of NF-kB. Although I think that it is a well-developed and interesting study, it is necessary to consider the following contents.
(1) In Figs. 6 and 7, authors speculated that (R)-(-)-carvone may activate Nrf2 and consequently decrease the availability of CBP/p300 and NF-kB interaction and NF-kB acetylation by CBP/p300. However, whether (R)-(-)-carvone inhibits the interaction between NF-kB and CBP/p300 has only been indirectly investigated. Direct proof should be provided by immunoprecipitation as to whether (R)-(-)-carvone inhibits the interaction.
R: We do agree with the reviewer, but unfortunately, we cannot perform that experiment for two main reasons. First, we don’t routinely use immunoprecipitation assays, so we would have to set up and optimize the technique before running the actual experiment. Second, we would have to prepare new nuclear extracts, as the leftovers are not sufficient to perform this assay. As all this would take too long, we acknowledged the limitation in the discussion section (lines 467-469 of the revised manuscript).
(2) A model diagram of the molecular mechanism of (R)-(-)-carvone obtained in this paper would help the reader to better understand it.
R: We modified the graphical abstract to include more mechanistic details and included it as figure 8.
Reviewer 2 Report
The authors have examined the mechanisms underlying anti-inflammatory effects of (R)-(-)-carvone in RAW 264.7 macrophage cells. The studies appear to be well done and fairly comprehensive. The manuscript fits well within this Special Issue as well. However, the following should be considered to strengthen the manuscript:
The crosstalk mentioned in the title is not investigated in the study. Although the authors do state this is a ‘potential’ role, it may stretch the results a bit.
In several instances, authors state a ‘tendency’ or an effect that is not supported by the data. This should be clarified with 1) showing the actual statistics and/or 2) toning down or removing these interpretations. Examples include: Ac-NF-kB data, HO-1, and Nrf2.
Although a couple of proteins, in which gene transcription is regulated by NF-kB are presented, transcriptional activity of NF-kB was not measured and so these conclusions should be toned down. This may be particularly relevant given the very brief LPS exposure in some instances (as short as 5 min), which would make it unlikely for transcription and translation to elicit robust changes.
Intro:
The Introduction outlines prior work and some rationale for the current study. Expanding on the relevance and potential implications would be useful to strengthen the need or utility of the study.
Materials and Methods:
2.2 – More detail on cell extraction is needed, rather than only referencing a previous paper. That paper also does not state the lysis buffer or other details, referencing yet another paper.
Results:
Supplemental Figures show the full blots cropped for paper figures. Were the other 3 biological replicates run in a similar pattern?
Throughout the Results, authors switch back and forth between the control or LPS being the comparison group. This should be consistent to avoid confusion. Particularly, when presented in the same figure.
Figure 3A – Why is Control not depicted in the graph?
A major outcome is the inhibition of LPS-induced JNK1 (but not other MAPK or JNK2/3). Although non-significant, there does appear to be a trend for inhibition of the other JNK isoforms. Presenting the statistics would be useful, particularly since authors note other ‘tendencies’ in the manuscript.
Figure 2B – Was pERK increased by R-cav?
Figure 4B – The cytosolic and nuclear extracts use different reference, as noted above. They also denote statistical differences differently.*** represents different vs LPS (as noted on control in nuclear), whereas # is different vs control. In all other figures, the * is on the Control to denote different vs LPS (rather than how it is in cytoplasmic, with # on the LPS bar).
Figure 4 – A verification blot of isolation is important. Although authors show tubulin and lamin, both of these would be present in whole cell lysates as well. The absence of one another in each fractionated lysate is important to demonstrate the approach.
Figure 6A – The graph is confusing. Please represent these data to match the way all other data are presented throughout the manuscript.
P9, line 300 – This statement regarding Ac-NF-kB is inaccurate as there is no change with LPS here.
Figure 6B – It would be useful to evaluate SIRT1 activity in the presence of LPS as well. Was this performed? Could the authors comment?
P11, line 357 – Authors state several results as though they are different, but none of this is supported statistically (which they briefly mention later). Providing the statistics may be helpful, or at least allow the reader to see the full picture and decide for themselves how to interpret these data.
Author Response
Coimbra, December 30th, 2022
Dear Editor and Reviewers:
On behalf of all authors, I thank you for the constructive comments and suggestions and for the opportunity to revise our manuscript. We think that the changes introduced greatly improved the manuscript and thank you again for your contribution.
As detailed below, we made significant changes to the text and some figures and also included additional data as supplementary tables to make clear how we reached the conclusions presented.
We do hope that you find the new version suitable for publication in this special issue of Pharmaceutics and remain at your disposal for any further corrections.
Kind regards and our best wishes for the new year,
Alexandrina Ferreira Mendes
Response to Reviewer 2
We thank the reviewer for the positive comments and constructive suggestions. Please, find our responses beneath each question.
The authors have examined the mechanisms underlying anti-inflammatory effects of (R)-(-)-carvone in RAW 264.7macrophage cells. The studies appear to be well done and fairly comprehensive. The manuscript fits well within this Special Issue as well. However, the following should be considered to strengthen the manuscript:
The crosstalk mentioned in the title is not investigated in the study. Although the authors do state this is a ‘potential’ role, it may stretch the results a bit.
R: We understand the reviewers point and changed the title to: Molecular mechanisms underlying the anti-inflammatory properties of (R)-(-)-carvone: potential roles of JNK1, Nrf2 and NF-kB which still conveys the idea that the mechanisms proposed are not definitely proven and does not imply any relation among the three proteins.
In several instances, authors state a ‘tendency’ or an effect that is not supported by the data. This should be clarified with 1) showing the actual statistics and/or 2) toning down or removing these interpretations. Examples include: Ac-NF-kB data, HO-1,and Nrf2.
R: Indeed, the results as presented do not clearly show why we considered the existence of tendencies in the levels of those proteins. In most cases (Nrf2, HO-1), we did so because the differences between LPS-treated cells in the absence and presence of R-carvone are very large, but vary a lot in absolute value among experiments. The large dispersion of results precludes statistical differences to be reached, but we do think that mean increases of 3.6- and 35-fold, respectively, cannot be due to random events, especially since in all experiments large increases were observed. To make this clearer, we included the raw data obtained in each experiment as supplemental material which clearly show that in every experiment R-carvone caused large increases in the cellular levels of both proteins. In the case of Ac-NF-kB, the effect induced by LPS relative to the control is indeed small and not statistically significant, but again the results were consistent in all experiments. Moreover, R-carvone decreased Ac-NF-kB levels to those obtained in the control or even less in all experiments, suggesting that the effect induced by LPS although small is real. On the other hand, the formula used to calculate changes in Ac-NF-kB levels was probably not the best approach. To make it simpler and present the results in a more consistent way throughout the manuscript, we normalized the protein levels using just the loading control (lamin), without considering total NF-kB/p65 levels which might have introduced a bias since they change in response to LPS and are not affected by treatment with R-carvone. Nonetheless, this change did not affect the overall interpretation as Ac-NF-kB levels in LPS-treated cells remained slightly higher than those obtained in the control and in the presence of R-carvone the reduction is maintained. Again, we included a table with the raw data as supplemental material and included the explanations above in the description of these results.
Although a couple of proteins, in which gene transcription is regulated by NF-kB are presented, transcriptional activity of NF-kB was not measured and so these conclusions should be toned down. This may be particularly relevant given the very brief LPS exposure in some instances (as short as 5 min), which would make it unlikely for transcription and translation to elicit robust changes.
R: NF-kB activation through the canonical activation pathway can occur at quite different speeds depending on the stimulus considered. Strong pro-inflammatory stimuli, like LPS, tumor necrosis factor-α or interleukin-1β, upon binding to their respective receptors induce IkB-α phosphorylation and degradation, followed by NF-kB nuclear translocation very rapidly. Indeed, we [1,2] and many others [3–5] observed that just 5 min are required for phosphorylated levels of IkB-α to increase significantly and IkB-α degradation is complete around 30 minutes after treatment with such stimuli in different cell types. Once NF-KB activation is initiated, gene expression is induced and naturally mRNA synthesis and translation take longer periods to occur, but the initial steps in the activation pathway can no longer be detected after those short time periods. Thus, to evaluate specific steps in the NF-kB activation pathway, different incubation times with the inducing stimulus are required, as done in this work.
Intro:
The Introduction outlines prior work and some rationale for the current study. Expanding on the relevance and potential implications would be useful to strengthen the need or utility of the study.
R: Please, refer to the introduction to see the information added (lines 35-39 and 45-49).
Materials and Methods:
2.2 – More detail on cell extraction is needed, rather than only referencing a previous paper. That paper also does not state the lysis buffer or other details, referencing yet another paper.
R: We added a detailed description of the protocols used for extraction of total proteins to the corresponding methods section.
Results:
Supplemental Figures show the full blots cropped for paper figures. Were the other 3 biological replicates run in a similar pattern?
R: Yes. The images show the same as those of all the other replicates run, as stated in figure legends.
Throughout the Results, authors switch back and forth between the control or LPS being the comparison group. This should be consistent to avoid confusion. Particularly, when presented in the same figure.
R: In most figures, comparisons are made relative to LPS-treated cells. The exceptions are proteins that are decreased by LPS treatment, namely IkBα and p65 in cytoplasmic extracts, or proteins on which the effect of LPS was uncertain (SIRT1, Nrf2 and HO-1). In the first case, if LPS was used as the comparator for those proteins, the graphs would appear quite odd as the value 1 would correspond to barely detectable or even undetectable bands.
Figure 3A – Why is Control not depicted in the graph?
We didn’t include the control in the graph in fig. 3A because the bands are barely detectable, but we agree that including it would be more informative and clearer. We now quantified that band in the 5 blots run and included the corresponding bar in the graph.
A major outcome is the inhibition of LPS-induced JNK1 (but not other MAPK or JNK2/3). Although non-significant, there does appear to be a trend for inhibition of the other JNK isoforms. Presenting the statistics would be useful, particularly since authors note other ‘tendencies’ in the manuscript.
R: We agree with the reviewer in that there is a tendency for reduced activation of JNK2 and 3, but the magnitude of the effect is small (less than 50%). So, we didn’t consider it to be a relevant tendency. For clarity and consistency, we added the statement “although a tendency to reduced levels of the activated forms can be observed“ to lines 185-186 of the revised manuscript and included the raw data as supplementary tables.
Figure 2B – Was pERK increased by R-cav?
R: We didn’t consider that increase to represent a relevant tendency for the same reason explained above. Indeed, unlike the very large increases observed for Nrf2 and HO-1, the increase in p-ERK levels is less than 50% of that observed in LPS-treated cells. However, unlike for JNK2 and 3, valuing that small increase opens the possibility that R-carvone can exert other effects related to ERK1/2 activation, namely increased cell survival [6]. Since such effect is out of the scope of this manuscript, we think that acknowledging the possibility that R-carvone can increase ERK activation would deviate the focus of the manuscript and make it too complex and probably speculative.
Figure 4B – The cytosolic and nuclear extracts use different reference, as noted above. They also denote statistical differences differently.*** represents different vs LPS (as noted on control in nuclear), whereas # is different vs control. In all other figures, the * is on the Control to denote different vs LPS (rather than how it is in cytoplasmic, with # on the LPS bar).
R: In fig. 4B, the comparison is made with reference to the control, since LPS treatment decreases p65 levels in the cytoplasm, as explained above. Thus, the symbol used to show statistically significant differences relative to the control was #. Since the results obtained in LPS-treated cells are statistically different from those obtained in the control, the symbol # was used above the LPS bar to show that.
Figure 4 – A verification blot of isolation is important. Although authors show tubulin and lamin, both of these would be present in whole cell lysates as well. The absence of one another in each fractionated lysate is important to demonstrate the approach.
R: We agree that cross incubation of each membrane with the antibody to the protein used as loading control in the other type of cell extract would be the ideal confirmation. However, in practice, it’s impossible to obtain completely pure cytoplasmic or nuclear extracts. Instead, we used tubulin or lamin as loading controls for cytoplasmic and nuclear extracts, respectively, to ensure that each extract is enriched in either cytoplasmic or nuclear proteins and not to demonstrate that they are completely pure. Corroborating this, in all cases the results show clear differences in the amount of the protein of interest (NF-kB/p65, Nrf2) in the extracts of control and LPS-treated cells, indicating that detection of the protein is not due to co-extraction of cytoplasmic or nuclear components.
Figure 6A – The graph is confusing. Please represent these data to match the way all other data are presented throughout the manuscript.
R: As mentioned above, the formula used to calculate Ac-NF-kB levels was probably not the best approach. To make it simpler and present the results in a more consistent way throughout the manuscript, we normalized the protein levels using just the loading control (lamin), without considering total NF-kB/p65 levels which might have introduced a bias since they change in response to LPS and are not affected by treatment with R-carvone. Nonetheless, this change did not affect the overall interpretation as Ac-NF-kB/p65 levels in LPS-treated cells remained slightly higher than those obtained in the control and in the presence of R-carvone the reduction, even below control levels, is maintained.
P9, line 300 – This statement regarding Ac-NF-kB is inaccurate as there is no change with LPS here.
R: The following sentence (line 301) mentioned explicitly that the differences are not statistically significant, but we recognize that it was unclear and changed it accordingly (lines 285-290 of the revised manuscript).
Figure 6B – It would be useful to evaluate SIRT1 activity in the presence of LPS as well. Was this performed? Could the authors comment?
R: The SIRT1 activity assay measures direct interaction between chemical compounds and the enzyme, allowing identification of both activators and inhibitors of its enzyme activity. The cellular effects of LPS are mediated by activation of its membrane receptor, TLR4. Thus, LPS doesn’t enter the cell and cannot physically interact with SIRT1. So, we didn’t include LPS in the “in tube” assay of SIRT1 activity. Nonetheless, since it was reported that in MC3T3 fibroblasts, LPS decreases SIRT1 protein levels [7], we used LPS to assess whether the anti-inflammatory effect of R-carvone in macrophages could be due to inhibition of LPS-induced downregulation of SIRT1 protein levels which was not observed.
P11, line 357 – Authors state several results as though they are different, but none of this is supported statistically (which they briefly mention later). Providing the statistics may be helpful, or at least allow the reader to see the full picture and decide for themselves how to interpret these data.
R: We agree with the reviewer that lack of statistical significance is a major limitation of this study that we could not overcome. As mentioned above, we included the raw data obtained in each experiment (Fig. 2C, Fig. 6A and Fig. 7) as supplementary tables to clearly show the consistency of the results obtained in replicate experiments and the magnitude of the effects induced by LPS and by R-carvone. We also included this limitation in the discussion (lines 456-459). We hope that, if the manuscript is accepted for publication, this will allow readers to unequivocally assess and acknowledge the relevance of the results obtained.
References of articles cited in this document:
- Rufino, A.T.; Ribeiro, M.; Judas, F.; Salgueiro, L.; Lopes, M.C.; Cavaleiro, C.; Mendes, A.F. Anti-Inflammatory and Chondroprotective Activity of (+)-α-Pinene: Structural and Enantiomeric Selectivity. J. Nat. Prod. 2014, 77, 264–269, doi:10.1021/np400828x.
- Rufino, A.T.; Ribeiro, M.; Sousa, C.; Judas, F.; Salgueiro, L.; Cavaleiro, C.; Mendes, A.F. Evaluation of the Anti-Inflammatory, Anti-Catabolic and pro-Anabolic Effects of E-Caryophyllene, Myrcene and Limonene in a Cell Model of Osteoarthritis. Eur. J. Pharmacol. 2015, 750, 141–150, doi:10.1016/j.ejphar.2015.01.018.
- Ernst, O.; Vayttaden, S.J.; Fraser, I.D.C. Measurement of NF-ΚB Activation in TLR-Activated Macrophages. In; 2018; pp. 67–78.
- Friedrichsen, S.; Harper, C. V.; Semprini, S.; Wilding, M.; Adamson, A.D.; Spiller, D.G.; Nelson, G.; Mullins, J.J.; White, M.R.H.; Davis, J.R.E. Tumor Necrosis Factor-α Activates the Human Prolactin Gene Promoter via Nuclear Factor-ΚB Signaling. Endocrinology 2006, 147, 773–781, doi:10.1210/en.2005-0967.
- Tse, A.K.-W.; Wan, C.-K.; Zhu, G.-Y.; Shen, X.-L.; Cheung, H.-Y.; Yang, M.; Fong, W.-F. Magnolol Suppresses NF-ΚB Activation and NF-ΚB Regulated Gene Expression through Inhibition of IkappaB Kinase Activation. Mol. Immunol. 2007, 44, 2647–2658, doi:10.1016/j.molimm.2006.12.004.
- Lu, Z.; Xu, S. ERK1/2 MAP Kinases in Cell Survival and Apoptosis. IUBMB Life (International Union Biochem. Mol. Biol. Life) 2006, 58, 621–631, doi:10.1080/15216540600957438.
- Ma, J.; Wang, Z.; Zhao, J.; Miao, W.; Ye, T.; Chen, A. Resveratrol Attenuates Lipopolysaccharides (LPS)-Induced Inhibition of Osteoblast Differentiation in MC3T3-E1 Cells. Med. Sci. Monit. 2018, 24, 2045–2052, doi:10.12659/MSM.905703.
Reviewer 3 Report
Reviewer’s Comments
The authors examined the effects of (R)-(-)-carvone on signaling pathways including MAPKs and NF-kB in the Raw 264.7 cell line to investigate the mechanism of its anti-inflammatory effects. (R)-(-)-carvone significantly inhibited LPS-induced JNK1 phosphorylation and IkB-α resynthesis. In addition, (R)-(-)-carvone showed a tendency to decrease nuclear acetylated NF-kB/p65 levels and to increase Nrf2 protein levels and HO-1 expression. These results suggest that JNK1 inhibition and Nrf2 activation by (R)-(-)-carvone may underlie its effects on NF-kB transcriptional activity and suppression of its target gene expression.
The manuscript should be of interest not only to mint researchers but also to a wide range of readers of the IJMS, since the experimental methods used to investigate the mechanism of the anti-inflammatory action of (R)-(-)-carvone are carefully planned, the steady experimental results are obtained, and the discussion based on these results is well presented. However, I think the following minor revisions are needed:
1. Since the signaling pathway involving MAPKs and NF-kB is very complex, it would be easier for readers to understand if a figure summarizing the mechanism of anti-inflammatory action of the putative (R)-(-)-carvone is inserted in the manuscript.
2. (R)-(-)-carvone may be useful for inflammation-related diseases such as osteoarthritis, so please include a discussion of its potential clinical applications, including in vivo experiments.
Author Response
Coimbra, December 30th, 2022
Dear Editor and Reviewers:
On behalf of all authors, I thank you for the constructive comments and suggestions and for the opportunity to revise our manuscript. We think that the changes introduced greatly improved the manuscript and thank you again for your contribution.
As detailed below, we made significant changes to the text and some figures and also included additional data as supplementary tables to make clear how we reached the conclusions presented.
We do hope that you find the new version suitable for publication in this special issue of Pharmaceutics and remain at your disposal for any further corrections.
Kind regards and our best wishes for the new year,
Alexandrina Ferreira Mendes
Response to Reviewer 3
We thank the reviewer for the positive comments and constructive suggestions. Please, find our responses beneath each question.
The authors examined the effects of (R)-(-)-carvone on signaling pathways including MAPKs and NF-kB in the Raw 264.7 cell line to investigate the mechanism of its anti-inflammatory effects.(R)-(-)-carvone significantly inhibited LPS-induced JNK1phosphorylation and IkB-α resynthesis. In addition, (R)-(-)-carvone showed a tendency to decrease nuclear acetylated NF-kB/p65 levels and to increase Nrf2 protein levels and HO-1expression. These results suggest that JNK1 inhibition and Nrf2activation by (R)-(-)-carvone may underlie its effects on NF-kB transcriptional activity and suppression of its target gene expression.
The manuscript should be of interest not only to mint researchers but also to a wide range of readers of the IJMS since the experimental methods used to investigate the mechanism of the anti-inflammatory action of (R)-(-)-carvone are carefully planned, the steady experimental results are obtained and the discussion based on these results is well presented. However, I think the following minor revisions are needed:
- Since the signaling pathway involving MAPKs and NF-kB is very complex, it would be easier for readers to understand if a figure summarizing the mechanism of anti-inflammatory action of the putative (R)-(-)-carvone is inserted in the manuscript.
R: We modified the graphical abstract to include more mechanistic details and included it as figure 8.
- (R)-(-)-carvone may be useful for inflammation-related diseases such as osteoarthritis, so please include a discussion of its potential clinical applications, including in vivo experiments.
R: Information regarding the therapeutic potential of anti-inflammatory agents, including monoterpenes, was inserted in the introduction (lines 35-39 and 45-49 of the revised manuscript). More specific information relative to R-carvone was added to the discussion (lines 430-455 of the revised manuscript).
Reviewer 4 Report
pharmaceutics-2106963
Abstract:
line 16 please include the definition of JNK1
line 17 please include the definition of Ik B-a
Line 75 please Escherichia coli change in italics
Line 97 please Tubulin change as tubulin; Lamin change as lamin
Author Response
Coimbra, December 30th, 2022
Dear Editor and Reviewers:
On behalf of all authors, I thank you for the constructive comments and suggestions and for the opportunity to revise our manuscript. We think that the changes introduced greatly improved the manuscript and thank you again for your contribution.
As detailed below, we made significant changes to the text and some figures and also included additional data as supplementary tables to make clear how we reached the conclusions presented.
We do hope that you find the new version suitable for publication in this special issue of Pharmaceutics and remain at your disposal for any further corrections.
Kind regards and our best wishes for the new year,
Alexandrina Ferreira Mendes
Response to Reviewer 4
We thank the reviewer for the positive evaluation. All corrections indicated were made to the corresponding lines in the revised manuscript.
Round 2
Reviewer 1 Report
While I believe that direct proof of the interaction is needed, we appreciate that the authors have added to the discussion the necessary considerations for the future.